# Air pollution, service development and innovation: Evidence from China

**Yong You** *

School of Economics, Zhejiang University, Hangzhou, China

* youyong@zju.edu.cn

## Abstract

The environment is crucial in economic development. However, the impact of the environment on cities, especially, on urban innovation still needs more attention. Based on city-level and $PM_{2.5}$ data from 280 prefecture-level cities and municipalities in China from 2003 to 2016, we empirically studied the influence of haze on urban innovation and its mechanism. The results show that (1) haze has a significant negative impact on urban innovation, which varies with regions, resource-dependent cities, provincial capital cities, and sub-provincial cities. (2) This result appears to be driven by industrial structure. By influencing the development of the service industry, haze affects the level of urban innovation. (3) In particular, advanced producer services and basic producer services are heterogeneous. Advanced producer services have a stronger impact on innovation, while basic producer services are more sensitive to haze. We used the Instrumental Variable (IV) regression to address endogeneity, the conclusion remains robust.

**Data Availability Statement:** All relevant data are within the manuscript and its Supporting Information files.

**Funding:** The author(s) received no specific funding for this work.

## 1. Introduction

The impact of urbanization, industrial structure, environmental regulation and environmental protection policies on air pollution attracts a lot of attention [1–3]. However, researchers have focused on pollution and economic activities fail to reach a consistent conclusion. Taking urbanization as an example, Shao et al. (2019) [1–3] find a significant positive correlation between urbanization and air pollution, while Wang (2011) [4] suggests a negative. Additionally, some studies reveal the "inverted U" or even more complex relationships [5, 6]. Since the relationship between economic activity and air pollution depends on the stage of development and industrial structure, these conflicting conclusions become reasonable. In the process of industrialization, economic activity and air pollution tend to have an "inverted U" relationship, which is the "environmental Kuznets curve" [7].

During the process of industrialization and urbanization, traffic emissions and congestion effects exacerbated the air pollution problem in China [8–10]. Various "city diseases" have become obstacles to China's green development. Adverse effects of air quality deterioration are not limited to increasing production costs [11], reducing labor productivity [12, 13], increasing the incidence of diseases [14]. Urban-level pollution emissions affect economic development through urbanization and human capital accumulation. Air pollution is

**Competing interests:** The authors have declared that no competing interests exist.

considered a main environmental risk factor for human health, due to its contribution to the development of pulmonary, cardiovascular, and neurological diseases, and mortality. Exposure to fine particulate matter ($PM_{2.5}$) is associated with cardiopulmonary mortality, cerebrovascular events, and overall mortality. The principal source of $PM_{2.5}$ is vehicle emissions and industrial emissions. Haze pollution significantly reduces urban attractiveness and economic development, which affects urbanization [15]. Haze destroys human capital accumulation and inhibits innovation and economic development [13]. Based on monitoring data of the mobile population in China, Sun et al. (2019) [16] confirmed that air pollution has a significant negative effect on the employment location of the mobile population.

Some studies focus on the impact of air pollution on innovation and productivity. Using a sample of Chinese manufacturing enterprises from 1998 to 2007, Fu et al., (2017) [17] find that air pollution can significantly reduce the per capita output. Based on firm data, Li and Zhang (2019) [18] estimate the impact of air pollution on TFP by using the regression discontinuity design. It is indicated that air pollution can significantly reduce the productivity of enterprises by increasing the human health risk, crowding out R&D funds, resource mismatch and reducing labor supply. This kind of research provides us with an important reference, but these studies face enterprise self-selection and the "pollution paradise" effect challenge which means polluting industries will relocate to jurisdictions with less stringent environmental regulations. Although the overall number of moving enterprises is very small [19] and the cost of relocation may be much higher than the cost of environmental improvement, studies on specific industries have identified the phenomenon of "workers-following-firms" which means the location and environment attract talents and firms gather then, such as Song et al. (2016)'s [20] research on the Chinese IT industry. This finding means that the industrial structure may be the key to understanding the impact of air pollution on innovation.

Industrial structure has played a key role in the impact of the environment on productivity and innovation. The industrial structure will evolve with the development of the economy to higher added value, lower energy consumption and lower pollution [21]. In 2013, the services' added value exceeded that of the secondary industry for the first time in China. Unlike agriculture and manufacturing, which rely on specific elements, such as land, minerals and equipment, services are more dependent on "human capital", which is more flexible and mobile. Although economic interests remain central to population mobility and human capital migration, other factors, such as air quality, have an increasingly important role [22]. Urban economics studies suggest that "local quality" is an important factor in human capital migration. Labor would "vote with their feet" which means the labor would choose where to live. The location of "creative talent" with higher innovation ability is determined by "local quality" (quality of place) and a superior natural environment is one of the core elements of the quality of life [23–25]. This kind of research shows that the ecological environment has become an important factor affecting the development of the local economy, especially affecting industrial development and talent selection.

Our analysis contributes to several strands of the literature. First, former studies investigate the impact of economic development on haze pollution and the impact of haze pollution on productivity. Our research expands the existing literature by exploring the impact of haze pollution on innovation in China. In particular, we investigate the mechanism by which haze affects innovation through services. To the best of our knowledge, this research is the first attempt in exploring haze's effect on China's innovation. Second, it adds to the analysis by constructing the environmental and innovation data sets on the prefectural city level from 2003 to 2016. By using global $PM_{2.5}$ satellite grid data to calculate the annual $PM_{2.5}$ average concentration of prefectural cities, we match the urban statistics, the patent database of the China National Intellectual Property Administrator, the urban innovation index and $PM_{2.5}$. Third,

Third, we contribute to the literature by using climate variables as instrumental variables. This addresses the endogeneity problems caused by the bidirectional effects of economic development and haze pollution effectively. Only a few studies use the instrumental variable method to address endogeneity [26].

The rest of this paper is organized as follows: the second section introduces the empirical strategy and data processing; the third part discusses the impact and heterogeneity of haze on innovation; section 4 analyzes the mechanism of haze affecting innovation; the fifth part comprises the robustness test; and the sixth part presents a summary of the full text.

## 2. Empirical strategies

### 2.1. Benchmark model and variables

In view of our research, we construct econometric models to investigate the impact of haze on innovation as follows:

$$\text{lninnov}_{it} = \theta_0 + \theta_1 \text{pm25} + \theta_2 \text{Control}_{it} + \epsilon_{it} \qquad (1)$$

In the equation, i represents the prefecture-level city sample, t represents the year, $\epsilon$ is the random disturbance term, and $\theta_1$, $\theta_2$, and $\theta_3$ are the parameters that remain to be estimated. lninnov is the explained variable innovation, which is measured by the logarithm of Kou and Liu (2017) urban innovation index [27]. The innovation index is based on patent renewal model which is proposed by Pake and Schankerman's (1984) [28] to evaluate patents' value. Kou and Liu (2017) [27] calculate the value of all the patents, and then add them up into cities. Compared with the number of patents, the urban innovation index values the patents and avoids the influence of "irrigation" on the number of patents. In the robustness test of the following text, we will also apply the number of invention patent authorizations as the explained variable for regression. The core explanatory variable is $PM_{2.5}$, which is employed to measure haze pollution. Existing studies usually utilize the real-time monitoring data published by the Ministry of Ecology and Environment of China, which are collected by observation stations. However, the historical information of these data is lacking. It was not until 2012 that $PM_{2.5}$ was included in official daily environmental monitoring. Therefore, we refer to Van Donkelaar (2015, 2019) [29, 30] and use satellite remote sensing data of global $PM_{2.5}$ which makes it possible to investigate the impact of air pollution on innovation in a long time dimension. These data have been applied in frontier studies in recent years [19].

Control represents other control variables that may affect innovation at the city level. Referring to existing research, we control eight variables: the scale of the city, economic development level, industrial structure, financial environment, economic development level, urbanization rate, Internet infrastructure and traffic infrastructure.

1. We use lnpop (log of population) to control the impact of city size on innovation;

2. we use lnpgdp (log of per capita GDP) to control the level of economic development. To avoid the disturbance of price factors, we refer to Zhang et al. (2004) [31], taking 2003 as the base period to deflate the GDP;

3. we control the urban financial environment using lnfina (log of per loan balance from financial institutions);

4. we use lnfdi (log of FDI) to control the openness of cities;

5. net (ratio of Internet users to the city's population) is employed to control Internet infrastructure (or information level);

6. lnroad (log of road construction area) is utilized to control traffic infrastructure;

7. we use urban (ratio of the district population to the city population) to control the rate of urbanization;

8. we use second (share of secondary industry employees) to control the urban industrial structure.

## 2.2. Data sources and data processing

Raw data are mainly derived from the Yearbook of Urban Statistics of China, China City and Industry innovation report of 2017, China National Intellectual Property Administration, global $PM_{2.5}$ satellite remote sensing data, and China Meteorological Database. Among them, the Yearbook of Urban Statistics of China reports the relevant variables of cities above the prefectural level in China; it is the main data source of urban-level control variables. There are some missing statistics for some cities. The treatment are as follows: (1) Delete the urban samples with more than three years and more than three missing control variables; (2) For missing variables in individual years and incorrect variables, we query a yearbook of provincial city statistics to correct the variables; (3) If data can not be obtained from other resources, then use the average value of the prior year data and subsequent year data; (4) The lack of FDI statistics exists only in some prefecture-level cities in the west provinces. We check its historical data. The missing value should be 0.

The China City and Industry Innovation report 2017 estimates the value of invention patents based on Pake and Schankerman's (1984) [28] patent renewal model and then divides the patent value into cities to build an urban innovation index. The value of the urban innovation index is more accurate and reasonable than the number of patents, so it is used as the explanatory variable. The global $PM_{2.5}$ satellite remote sensing data are longitude and latitude raster data. We use ArcGIS to locate the $PM_{2.5}$ in each prefecture-level city and then calculate the annual mean level of $PM_{2.5}$ concentration in each prefecture-level city. China Meteorological Data are meteorological data at the prefecture-level city level, including annual sunshine hours, precipitation and average temperature. In addition, the invention patent date from the China National Intellectual Property Administration is summed to the city level as the explanatory variable in the robustness test. These city data are matched with the administrative code or city name. The balanced panel data of 280 cities above the prefectural level from 2003 to 2016 are constructed in this paper. The descriptive statistics of the data are shown in Table 1.

**Table 1. Descriptive statistics.**

| variable | meaning | name | sample | mean | sd | min | max |
|---|---|---|---|---|---|---|---|
| explained variable | log of urban innovation index | lninnov | 3920 | -0.301 | 1.900 | -4.605 | 6.967 |
| explanatory variable | $PM_{2.5}$ | pm25 | 3920 | 0.445 | 0.193 | 0.031 | 1.101 |
| control variable | log of per loan balance from financial institutions | lnfina | 3920 | 15.862 | 1.357 | 12.548 | 21.007 |
| | log of population | lnpop | 3920 | 5.861 | 0.698 | 2.795 | 9.315 |
| | share of secondary industry employees | second | 3920 | 0.420 | 0.162 | -0.786 | 0.988 |
| | ratio of the district population to the city population | urban | 3920 | 0.346 | 0.238 | 0.034 | 1.000 |
| | log(FDI+1) | lnfdi | 3920 | 9.146 | 2.775 | 0.000 | 14.941 |
| | log of per capita GDP | lnpgdp | 3920 | -5.551 | 0.711 | -8.879 | -2.516 |
| | ratio of Internet users to the city's population | net | 3920 | 0.117 | 0.128 | 0.000 | 0.766 |
| | log of road construction area | lnroad | 3,920 | -2.617 | 1.420 | -7.042 | 1.864 |

## 2.3. Estimation methods

We use eight control variables to control the impact of other factors on innovation. However, considering that there may still be other unobservable variables (such as cultural atmosphere, innovation environment, etc.), the estimation of the ordinary least squares (OLS) method is probably biased and inconsistent. The choice of the fixed effect (FE) model can solve the influence of unobservable variables at the city level. Since we use 2003–2016 as the investigation period, the time effect of such a long time should also be considered, so we choose a two-way fixed effect model. To address the heteroscedasticity caused by other possible missing variables, the heteroscedasticity-robust standard error is used to control the heteroscedasticity in regression. In addition, as an important economic variable, innovation and pollution may have two-way causality, which makes the conclusion face the challenge of endogeneity. To solve this problem, this paper uses instrumental variable regression (IV) to further test the robustness, as detailed in the following section.

## 3. Relationship between haze and innovation

This section estimates the average impact of haze on urban innovation and then discusses the heterogeneity of haze on urban innovation from three dimensions: regional differences between east and west, key cities and non-key cities, and resource-dependent cities and non-resource-dependent cities.

### 3.1. Baseline model

Table 2 reports the regression results of the benchmark regression model. Columns (1) and (2) present the OLS regression results. The difference is whether to control the year fixed effect. Whether time-fixed effects were controlled, the pm25 coefficient was significantly negative at the significance level of 5%. The coefficient symbols of the explanatory variables in the fixed effect model regression are consistent with OLS regression, as shown in column (3) and column (4), respectively. This finding verifies the negative relationship between haze pollution and innovation. In column (5), a quadratic term of pm25 is added to the equation to investigate whether there is a complex linear relationship similar to the "environmental Kuznets curve" between haze and innovation. As shown in column (5), the coefficient of pm25 is still

**Table 2. Baseline model.**

|  | (1) | (2) | (3) | (4) | (5) |
|---|---|---|---|---|---|
|  | OLS | OLS | FE | FE | FE |
| pm25 | -0.574*** | -0.250*** | -2.091*** | -0.707*** | -2.781*** |
|  | (0.072) | (0.069) | (0.151) | (0.162) | (0.471) |
| $(pm25)^2$ |  |  |  |  | 1.872*** |
|  |  |  |  |  | (0.355) |
| Control | no | yes | yes | yes | yes |
| Year | no | yes | no | yes | yes |
| FE | no | no | yes | yes | yes |
| N | 3920 | 3920 | 3920 | 3920 | 3920 |
| adj. $R^2$ | 0.873 | 0.885 | 0.869 | 0.896 | 0.898[1] |

[1] Note

*, **, *** represent significant at 10%, 5% and 1% significant levels, respectively. FE represent city fixed effects. Year means year fixed effects. Regression uses heteroscedasticity robust standard errors. Values below coefficients are standard errors. Constant term coefficient is not reported.

negative, but the quadratic coefficient is positive, which seems to indicate a "U type" relationship between haze and innovation.

How can the U-shaped relationship between haze and innovation be understood? We suggest that this relationship reflects the gradual change in economic development. With a focus on the environment and the progress of technology, the means of production are transformed from high pollution to an environmentally friendly mode. From the data, the inflection point of the U-type relationship calculated by the regression coefficient, where $PM_{2.5} = 0.74$. Combined with descriptive statistics, 93% of the samples' $PM_{2.5}$ was below 0.74, which means that for most cities, the relationship between haze and innovation is one-way. The cities above the critical value are mostly northern cities with developed heavy industry (such as Tianjin, Shijiazhuang, Langfang, etc.). From the dimension of time, according to the Kernel Density Estimation (KDE) of $PM_{2.5}$ which is reflected in the Fig 1, year 2012 is a watershed for haze trends. Before 2012, haze deteriorated overall, the average haze concentration continued to rise, and the high pollution city density increased. However, starting in 2012, as the public paid more attention to $PM_{2.5}$, the Ministry of Ecology and Environment of China included $PM_{2.5}$ in its routine monitoring for the first time. Local ecological control and anti-haze action began to achieve their goals. The level of haze at the national level has gradually decreased. The average haze concentration shifts to the left, and the proportion of high-pollution cities is also

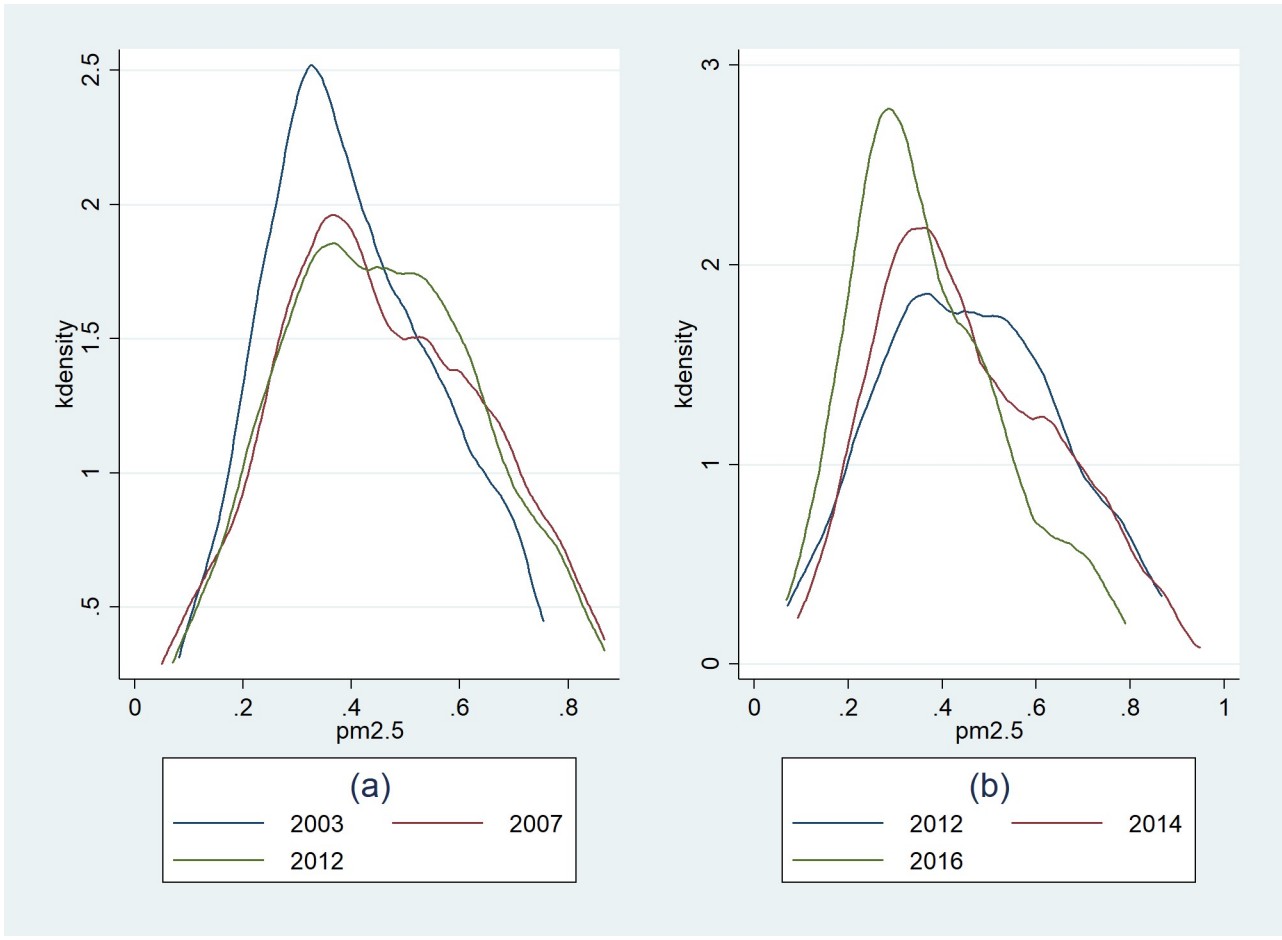

**Fig 1.** Kernel Density Estimation of $PM_{2.5}$ concentration: (**a**) Description of $PM_{2.5}$ before 2012; (**b**) Description of $PM_{2.5}$ after 2012.

declining. Predictably, in the future, haze concentrations at the national level will continue to shift to the left., Haze would only have a negative impact on innovation one day.

## 3.2. Heterogeneity analysis

Benchmark regression verifies the relationship between haze and urban innovation. However, what factors lead to the impact of haze on innovation? Existing research on Chinese innovation finds that urban innovation is related to regional characteristics, urban endowment, industrial structure and other factors. Do those factors lead to differences in the impact of haze on urban innovation?

At the regional level, there are differences in the market environment and cultural concepts in the eastern, central and western regions. Table 3 reports the differences in the impact of haze on innovation in the eastern, central and western regions. All cities were divided into three regions according to provinces. Columns (2), (3) and (4) are the east, central and west subsample regression, respectively. The regression coefficient shows that the influence of haze on innovation is heterogeneous. Air pollution seems to have the greatest effect on innovation in the east, while the coefficient of air pollution in the western region is minimally significant. This result is consistent with our expectations. The resource endowments of the central and western regions are higher than those of the eastern region, especially the rich resources of the provinces in some parts of the western region (such as Shaanxi, Shanxi, Guizhou, etc.), which means high energy dependence. On the other hand, the pollution-related heavy industry in the central and western regions are more developed and more dependent on these industries. Thus, the reduction in haze has a limited impact on innovation. In contrast, innovation in the east is more sensitive to air pollution.

There are two kinds of endowments: natural resource endowments and policy endowments. Although natural resources give cities comparative advantages in the process of development, they even make Ordos the city with the highest per capita GDP in China. Resource-dependent cities lack incentives for innovation for a long time because they rely too much on the natural resources industry, which is not conducive to industrial transformation and upgrading and may force themselves to fall into the "resource trap" in which countries or areas with an abundance of non-renewable natural resources experience stagnant economic growth or even economic contraction. In China, policy resources are a key resource in city development. Policy endowment will generate continuous resource investment, which would not make urban development fall into the "resource trap", but the dependence on the environment is low because there are other irreplaceable advantages that affect innovation.

**Table 3. Heterogeneity.**

|  | (1) | (2) | (3) | (4) |
|---|---|---|---|---|
|  | baseline | east | contral | west |
| pm25 | -0.707*** | -0.899*** | -0.646*** | -0.312 |
|  | (0.162) | (0.304) | (0.235) | (0.320) |
| Control | yes | yes | yes | yes |
| Year | yes | yes | yes | yes |
| FE | yes | yes | yes | yes |
| $N$ | 3920 | 1386 | 1386 | 1148 |
| adj. $R^2$ | 0.896 | 0.927 | 0.883 | 0.896 |

Note

*, **, *** represent significant at 10%, 5% and 1% significant levels, respectively. FE represent city fixed effects. Year means year fixed effects. Regression uses heteroscedasticity robust standard errors. Values below coefficients are standard errors. Constant term coefficient is not reported.

Therefore, we compare resource-dependent cities with non-resource-dependent cities and key cities (provincial capital cities and sub-provincial cities) with non-key cities. The list of resource-dependent cities is published by the State Council of China, in which the resource cities are rich in natural resources as oil, coal, ore and wood. Considering that wood is a renewable resource, we exclude 3 forest cities from all the 66 resource cities in the list. Because of resource advantage, the mining and metallurgy in those cities are well developed. While the key cities are provincial capital cities and sub-provincial cities which usually have a long history and a special political status.

Table 3 reports the regression results of the endowment differences. Column (1) is listed as the benchmark regression and columns (2) and (3) are listed as the regression of non-resource cities and resource cities. The results show that the absolute value of the air pollution coefficient is low. The absolute value of the pollution coefficient is 0.68, which is lower than that of the non-resource city (0.71). It is not difficult to understand that resource-dependent cities are cities with mining and processing of natural resources, such as minerals in the region as the leading industries. The industrial level has a strong dependence on resources, high energy consumption, high pollution, and high emission projects (mining and washing of coal, processing of ferrous metalsores, processing of nonferrousmetals ores, oli processing and coking, etc.), hindering the development of alternative industries. The attraction of talent and funds is also insufficient. Therefore, haze and other air pollution factors have a relatively small impact on innovation.

Columns (4) and (5) of Table 4 report the regression results of key cities and non-key cities, respectively. Key cities are provincial capitals and sub-provincial cities in China, and the rest are non-key cities. The regression results show that the influence of air pollution on the innovation of key cities is smaller, the coefficient is -0.51, whose absolute value is less than that of -0.67. We speculate that this result may be due to other irreplaceable locations and policy and resource advantages of key cities, resulting in a lesser impact of air pollution on their innovations than that of ordinary cities.

## 4. Mechanism analysis

Through the analysis of influence differences, this paper confirms that the influence of haze on innovation varies with regional and endowment differences. Further analysis seems to point to its relationship with industrial structure, which naturally leads to the focus of this paper on industrial structure differences. Li and Zhang (2019) [18] found that air pollution has caused the productivity of heavily polluted industries to rapidly decline. These heavy industries and

**Table 4. Baseline model.**

|  | (1) | (2) | (3) | (4) | (5) |
|---|---|---|---|---|---|
|  | baseline | non-resource-dependent cities | resource-dependent cities | key cities | non-key cities |
| pm25 | -0.707*** | -0.706*** | -0.682** | -0.514* | -0.672*** |
|  | (0.162) | (0.197) | (0.292) | (0.290) | (0.174) |
| Control | yes | yes | yes | yes | yes |
| Year | yes | yes | yes | yes | yes |
| FE | yes | yes | yes | yes | yes |
| $N$ | 3920 | 3097 | 823 | 490 | 3430 |
| adj. $R^2$ | 0.896 | 0.893 | 0.913 | 0.960 | 0.891 |

Note

*, **, *** represent significant at 10%, 5% and 1% significant levels, respectively. FE represent city fixed effects. Year means year fixed effects. Regression uses heteroscedasticity robust standard errors. Values below coefficients are standard errors. Constant term coefficient is not reported.

manufacturing products produce more air pollution. Based on these considerations, we discuss industry heterogeneity and the impact mechanism of pollution on innovation.

## 4.1. Effects of industrial structure

Regarding industrial structure, there are two ways to depict existing research: the proportion of employees and the proportion of added value. In the Yearbook of Urban Statistics, the proportion of added value and the number of employees in the first, second and third industries are published; however, at a fine scale, each subindustry only provides employee number information. Because of the diversity of each subindustry, to facilitate further discussion, based on Gozgor (2018) [32] and Chen and Wang (2021) [2], the proportion of employees is used to measure the development of various industries. In the regression in Table 5, the first, second and third industry proportions (first, second, and third) and their cross terms with pm25 are added to this paper. Among them, (1) as a baseline regression, columns (2), (3), and (4) add the cross terms of the first, second, tertiary and haze, respectively. All cross terms are placed in column (5). Based on the regression results in column (5), haze has only interactive effects with the tertiary industry, that is, the more developed the tertiary industry is, the worse haze inhibits its innovation. This finding shows that the impact of haze on innovation is mainly caused by the tertiary industry. According to this article, this impact is mainly due to the dependence of agriculture, industry and other fixed elements, such as land and plant equipment. Services rely more on mobile human capital, and their employees are sensitive to air quality and other environments. The improvement of urban livability has attracted the entry of employees of human capital, especially the tertiary industry, and then optimizes the urban industrial structure and improves the innovation ability.

## 4.2. Mediation of services

The results of adding interaction items show that the influence of haze on innovation is related to the tertiary industry. This section further analyzes the impact. According to Baron & Kenny

**Table 5. Effects of industrial structure.**

|  | (1) | (2) | (3) | (4) | (5) |
|---|---|---|---|---|---|
|  | lninnov | lninnov | lninnov | lninnov | lninnov |
| pm25 | -0.680*** | -0.573*** | -1.064*** | -0.139 | -0.427 |
|  | (0.164) | (0.167) | (0.276) | (0.246) | (0.283) |
| first×pm25 |  | -0.038** |  |  | -0.024 |
|  |  | (0.016) |  |  | (0.017) |
| second×pm25 |  |  | 0.008 |  | 0.007 |
|  |  |  | (0.005) |  | (0.005) |
| third×pm25 |  |  |  | -0.010*** | -0.009*** |
|  |  |  |  | (0.003) | (0.003) |
| Control | yes | yes | yes | yes | yes |
| Year | yes | yes | yes | yes | yes |
| FE | yes | yes | yes | yes | yes |
| N | 3920 | 3920 | 3920 | 3916 | 3916 |
| adj. $R^2$ | 0.896 | 0.896 | 0.896 | 0.897 | 0.897 |

Note

*, **, *** represent significant at 10%, 5% and 1% significant levels, respectively. FE represent city fixed effects. Year means year fixed effects. Regression uses heteroscedasticity robust standard errors. Values below coefficients are standard errors. Constant term coefficient is not reported.

(1986) [33] and Wen and Ye (2014) [34], combined with the benchmark measurement model, this paper constructs the standard mediation effect test "three steps" as follows:

$$\text{lninnov}_{it} = \theta_0 + \theta_1 \text{pm25} + \theta_2 \text{Control}_{it} + \epsilon_{it} \tag{2}$$

$$\text{M}_{it} = \beta_0 + \beta_1 \text{pm25} + \beta_2 \text{Control}_{it} + \mu_{it} \tag{3}$$

$$\text{lninnov}_{it} = \gamma_0 + \gamma_1 \text{pm25} + \gamma_2 \text{M}_{it} + \gamma_3 \text{Control}_{it} + \tau_{it} \tag{4}$$

where Control is a set of control variables and M is an intermediary variable. If there is a "mediation effect", that is, the explanatory variable pm25 affects the explained variable via the M intermediary variable. The coefficients $\theta_1$, and $\beta_1$ would be significant, and the absolute value of $\gamma_1$ is smaller than that of $\theta_1$, or $\gamma_1$ is no longer significant. Table 6 reports the regression results, and columns (1), (2) and (3) are the regression results of Eqs (2), (3) and (4), respectively. In column (2), the coefficient of pm25 is highly significant, which means that haze will affect the development of services. The results of column (3) clearly show that after adding the mediation variable into the model, the absolute value of haze's coefficient on innovation (0.51) is lower than that of benchmark regression (0.71). Haze affects urban innovation via services development.

## 4.3. Further discussion about services

How does the service industry affect innovation? There are actually two kinds of services: a consumer service industry that directly meets final demand (consumer services) and producer services (producer services), which are employed as intermediate inputs by producers of other goods and services. Producer services include both small producer services (finance, insurance, law, etc.) and large distributive services (e.g., commerce, transport, communications, etc.), which contains a large amount of human capital and knowledge capital. Based on existing research, there are three main mechanisms influencing innovation in producer services: (1) R & D, design, marketing, consulting and other functional departments are separated and independent from manufacturing. This specialization promotes innovation in producer services [35]. (2) Upstream of manufacturing, producer services interact with manufacturing. The spillover effects of scale economies and technology would promote the efficiency of downstream

**Table 6. Mediation of services.**

|  | (1) | (2) | (3) |
|---|---|---|---|
|  | **lninnov** | **lnse** | **lninnov** |
| lnse |  |  | 0.597*** |
|  |  |  | (0.104) |
| pm25 | -0.707*** | -0.326*** | -0.509*** |
|  | (0.162) | (0.073) | (0.156) |
| Control | yes | yes | yes |
| Year | yes | yes | yes |
| FE | yes | yes | yes |
| N | 3920 | 3918 | 3918 |
| adj. $R^2$ | 0.896 | 0.566 | 0.903 |

Note

*, **, *** represent significant at 10%, 5% and 1% significant levels, respectively. FE represent city fixed effects. Year means year fixed effects. Regression uses heteroscedasticity robust standard errors. Values below coefficients are standard errors. Constant term coefficient is not reported.

manufacturing enterprises and upgrade the regional industrial structure [36]. (3) The agglomeration of producer services enhances human capital accumulation and increases opportunities for technical staff to communicate and cooperate with each other, thus creating a good learning and innovation environment [37]. By technology spillovers, enabling the diffusion and sharing of knowledge and information, producer services promote production, extend industrial value chains, improve industrial R&D, product design and scientific management, and enhance production efficiency in industrial enterprises. Recent studies have also found that air pollution affects the development of producer services. For example, Chen and Qian (2020) [38], which based on the "low carbon city" policy in China, empirically confirmed that environmental protection increased the proportion of producer services in the city. These studies suggest that producer services are likely to be an important mechanism by which air pollution affects innovation behavior. Although the nonproducer service industry will not directly affect enterprise innovation, it is also an important part of local quality or amenities under the framework of urban economics. The convenience of living provided by catering, accommodation, etc. is an important factor affecting the location choice of human capital and may also have an impact on innovation.

Referring to Xuan and Yu (2017) and Chen and Qian (2020) [38, 39], the service industry is divided into producer services (ps) and nonproducer services or consumer services (cs). We consider transport warehousing, information transmission, leasing services, scientific research services, and finance as producer services. Other nonproducer services include wholesale and retail, accommodation and catering, real estate, and maintenance services. We use the log of the employees to measure their development. Table 7 reports the results of the producer services and nonproducer services mediation regression. The explanatory variables listed in columns (2) and (3) are producer services and nonproducer services development. The coefficients of the core explanatory variables are significantly negative, which shows that haze has a significant negative impact on producer services and nonproducer services. After the mediator is added in column (4), the coefficient of the explanatory variable turns into -0.5. The absolute value is also lower than the value of 0.71 in the benchmark regression. According to the setting of the mediation model. Haze affects innovation by affecting producer services and nonproducer services.

**Table 7. Producer services (ps) and Consumer services (cs).**

|  | (1) | (2) | (3) | (4) |
|---|---|---|---|---|
|  | lninnov | lnps | lncs | lninnov |
| lnps |  |  |  | 0.231*** |
|  |  |  |  | (0.084) |
| lncs |  |  |  | 0.411*** |
|  |  |  |  | (0.117) |
| pm25 | -0.707*** | -0.301*** | -0.328*** | -0.500*** |
|  | (0.162) | (0.091) | (0.071) | (0.156) |
| Control | yes | yes | yes | yes |
| Year | yes | yes | yes | yes |
| FE | yes | yes | yes | yes |
| N | 3920 | 3918 | 3920 | 3918 |
| adj. $R^2$ | 0.896 | 0.479 | 0.551 | 0.903 |

Note

*, **, *** represent significant at 10%, 5% and 1% significant levels, respectively. FE represent city fixed effects. Year means year fixed effects. Regression uses heteroscedasticity robust standard errors. Values below coefficients are standard errors. Constant term coefficient is not reported.

**Table 8. Advanced producer services (aps) and basic producer services (bps).**

|  | (1) | (2) | (3) | (4) |
|---|---|---|---|---|
|  | **lninnov** | **lnaps** | **lnbps** | **lninnov** |
| lnaps |  |  |  | 0.338*** |
|  |  |  |  | (0.091) |
| lnbps |  |  |  | 0.184*** |
|  |  |  |  | (0.051) |
| pm25 | -0.707*** | -0.217*** | -0.357*** | -0.565*** |
|  | (0.162) | (0.071) | (0.134) | (0.153) |
| Control | yes | yes | yes | yes |
| Year | yes | yes | yes | yes |
| FE | yes | yes | yes | yes |
| N | 3920 | 3918 | 3920 | 3918 |
| adj. $R^2$ | 0.896 | 0.604 | 0.279 | 0.902 |

Note

*, **, *** represent significant at 10%, 5% and 1% significant levels, respectively. FE represent city fixed effects. Year means year fixed effects. Regression uses heteroscedasticity robust standard errors. Values below coefficients are standard errors. Constant term coefficient is not reported.

Furthermore, "transport warehousing" and "rental services" are regarded as basic producer services (bps), while "finance", "information transmission", and "finance" are considered advanced producer services (aps). We compare the mechanism through which haze affects urban innovation. Columns (2) and (3) of Table 8 are regressions that use advanced producer services development and basic producer services development, respectively, as explanatory variables. The coefficients of haze are significant. In column (4), as a result of joining two categories of producer services, the coefficient of explanatory variables is less than that of benchmark regression in column (1). This result proves that haze affects high and basic producer services and then affects urban innovation. In column (4), the regression coefficient of advanced producer services is higher than that of basic producer services, which means that advanced producer services have a stronger impact on innovation. Comparing the results of columns (2) and (3), the coefficients of pm25 are -0.22 and -0.36, which suggests that advanced producer services are less affected by haze. Although this finding seems to contradict the higher requirements of highly skilled human capital for amenities, we hold the view that advanced producer services, such as finance, scientific research, and information transmission, support the demand for local manufacturing, and development is affected by not only air pollution but also the local industrial structure, economic level and other factors. Therefore, compared with basic producer services, it is less insensitive to haze.

## 5. Robustness test

### 5.1. IV regression

The two-way fixed effect model and intermediary effect model are used to confirm the mechanism by which haze influences innovation through the service industry. Although the fixed effect model addresses some difficult-to-observe variables that control the city, air pollution is usually related to industrial emissions. Innovation and production efficiency improvement may affect pollution emissions and cause haze and innovation to have a two-way causality. Considering the endogeneity of this bidirectional causality, we use instrumental variable (IV) regression for further testing. The difficulty of IV lies in externality, that is, it is necessary to find indicators related to explanatory variable haze but not affected by urban innovation of

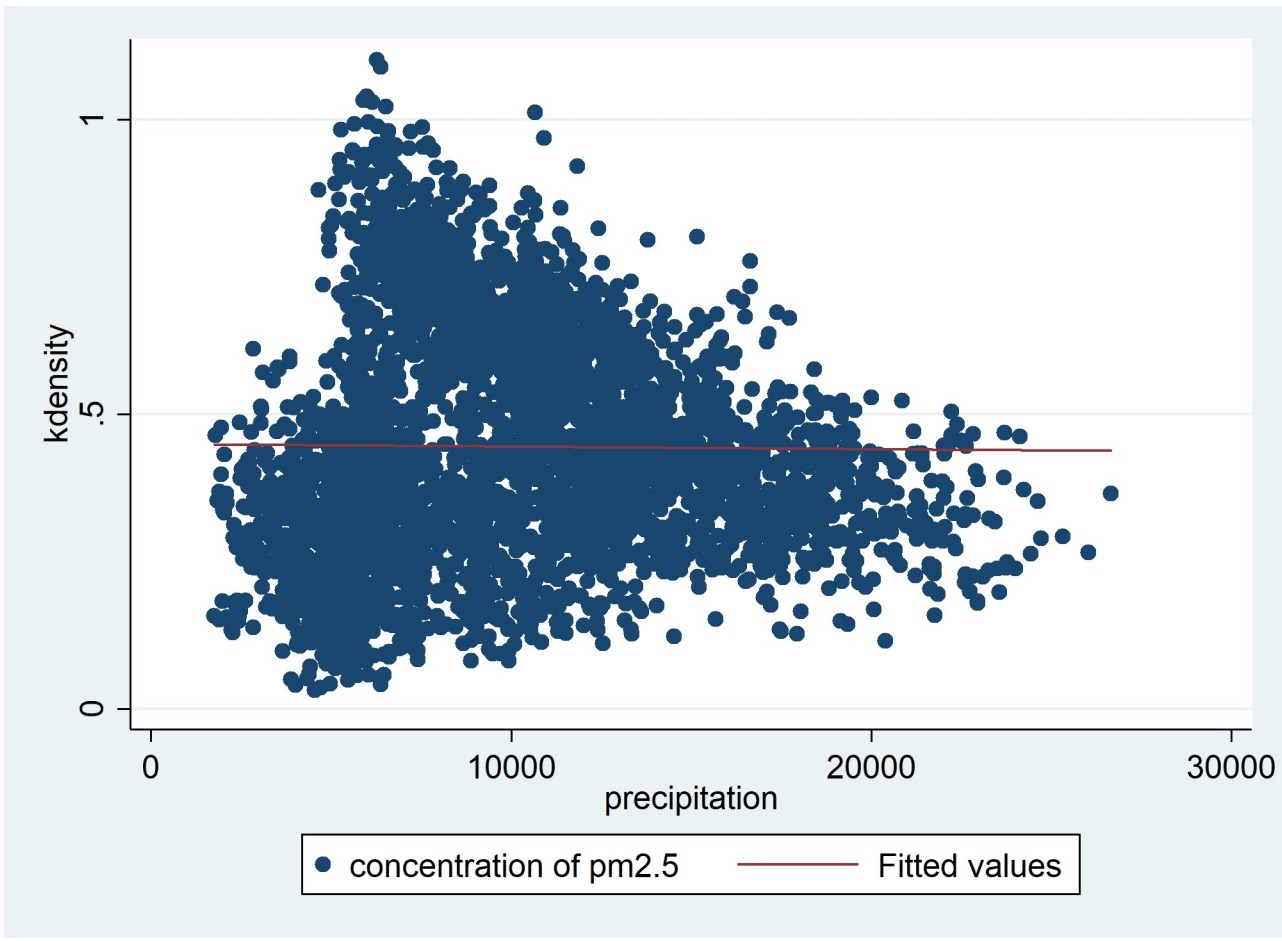

**Fig 2. The relationship between precipitation and haze concentration of cities.**

explained variables and the service industry in the intermediary effect model. Thus, bidirectional causality is interrupted. The formation of haze is the result of the interaction between human activities and natural factors (geography, climate, etc.). Human activities have economic and social attributes and may have endogenous relationships with innovation, while natural factors are relatively exogenous, so it is a good idea to use natural variables as IVs. The geographical terrain would not change over time, so it is not suitable as an IV for the fixed effect model. Therefore, climate variables, which not only have exogenous attributes but also affect the diffusion rate of pollutants and are related to haze, are chosen. In this paper, we choose the precipitation in the China Meteorological Database as an IV of haze. The logic is that the scour of rain water will freshen the air, and the haze concentration will decrease when precipitation increases. Fig 2 reflects the negative relationship between haze and precipitation.

Tables 9–11 are the IV regressions of the mechanism test. Consider Table 9; for example, the F value of the first stage is greater than 10, and the p value is 0, which rejects the weak instrumental variable hypothesis. The coefficient of regression in the first stage is significant at the 5% level, which indicates the effectiveness of the instrumental variables. The results of the two-stage regression report are similar to the previous mechanism analysis, which means that even if we exclude possible endogeneity, the conclusions are still valid. This conclusion further proves that haze affects innovation through the service industry mechanism. The conclusions in Tables 10 and 11 are also consistent with these findings in the mechanism analysis.

**Table 9. Mechanism analysis of IV regression.**

| | (1) | (2) | (3) |
|---|---|---|---|
| | **lninnov** | **lnse** | **lninnov** |
| lnse | | | 0.378*** |
| | | | (0.107) |
| pm25 | -7.290*** | -1.625*** | -6.667*** |
| | (1.622) | (0.490) | (1.648) |
| Control | yes | yes | yes |
| FE | yes | yes | yes |
| Year | yes | yes | yes |
| IV | -0.042*** | -0.042*** | -0.039*** |
| | (0.005) | (0.005) | (0.005) |
| F | 107.04 | 107.22 | 103.71 |
| N | 3920 | 3918 | 3918 |
| $R^2$ | 0.8107 | 0.4876 | 0.8288 |

Note

*, **, *** represent significant at 10%, 5% and 1% significant levels, respectively. FE represent city fixed effects. Year means year fixed effects. Regression uses heteroscedasticity robust standard errors. Values below coefficients are standard errors. Constant term coefficient is not reported. IV in the table means precipitation. The coefficient of IV and F value is in the first stage regression.

## 5.2. Measuring innovation using invention patents

To further demonstrate the robustness of this study, this section uses the log of invention patents to measure innovation to replace the urban innovation index as an explained variable.

**Table 10. Producer services and consumer services.**

| | (1) | (2) | (3) | (4) |
|---|---|---|---|---|
| | **lninnov** | **lnps** | **lncs** | **lninnov** |
| lnps | | | | 0.191* |
| | | | | (0.099) |
| lncs | | | | 0.215* |
| | | | | (0.122) |
| pm25 | -7.291*** | -1.323** | -1.615*** | -6.682*** |
| | (1.622) | (0.647) | (0.468) | (1.646) |
| Control | yes | yes | yes | yes |
| FE | yes | yes | yes | yes |
| Year | yes | yes | yes | yes |
| IV | -0.042*** | -0.042*** | -0.042*** | -0.039*** |
| | (0.005) | (0.005) | (0.005) | (0.005) |
| F | 107.04 | 107.22 | 107.04 | 97.60 |
| N | 3920 | 3918 | 3920 | 3918 |
| $R^2$ | 0.8107 | 0.4456 | 0.4764 | 0.8288 |

Note

*, **, *** represent significant at 10%, 5% and 1% significant levels, respectively. FE represent city fixed effects. Year means year fixed effects. Regression uses heteroscedasticity robust standard errors. Values below coefficients are standard errors. Constant term coefficient is not reported. IV in the table means precipitation. The coefficient of IV and F value is in the first stage regression.

**Table 11. Advanced producter services and basic producter services.**

|  | (1) | (2) | (3) | (4) |
|---|---|---|---|---|
|  | lninnov | lnaps | lnbps | lninnov |
| lnaps |  |  |  | 0.236** |
|  |  |  |  | (0.096) |
| lnbps |  |  |  | 0.131** |
|  |  |  |  | (0.057) |
| pm25 | -7.291*** | -1.918*** | -0.566 | -6.755*** |
|  | (1.622) | (0.622) | (0.876) | (1.593) |
| Control | yes | yes | yes | yes |
| FE | yes | yes | yes | yes |
| Year | yes | yes | yes | yes |
| IV | -0.042*** | -0.042*** | -0.042*** | -0.040*** |
|  | (0.005) | (0.005) | (0.005) | (0.005) |
| F | 107.04 | 107.22 | 107.04 | 100.43 |
| N | 3920 | 3918 | 3920 | 3918 |
| $R^2$ | 0.8107 | 0.4967 | 0.2820 | 0.8271 |

Note

*, **, *** represent significant at 10%, 5% and 1% significant levels, respectively. FE represent city fixed effects. Year means year fixed effects. Regression uses heteroscedasticity robust standard errors. Values below coefficients are standard errors. Constant term coefficient is not reported. IV in the table means precipitation. The coefficient of IV and F value is in the first stage regression.

The patent mainly includes invention patent, utility model and design. The invention patent needs to satisfy the standard of practicality, novelty and creativity. The other two kinds of patents only need to satisfy certain practicality and novelty [27]. The number of invention patents is usually used to measure innovation, especially valuable innovations. However, the number of patent applications is affected by local innovation incentive policies, such as patent subsidy policies, bubbles, and irrigation. Patent licensing depends on the approval of the State Intellectual Property Office [40]. Therefore, we use the log of the number of invention patents as the explanatory variable. The results are reported in Tables 12–14. After replacing the explained

**Table 12. Baseline model.**

|  | (1) | (2) | (3) |
|---|---|---|---|
|  | lnpatent | lnse | lnpatent |
| lnse |  |  | 0.617*** |
|  |  |  | (0.114) |
| pm25 | -0.596*** | -0.326*** | -0.394** |
|  | (0.196) | (0.073) | (0.194) |
| Control | yes | yes | yes |
| Year | yes | yes | yes |
| FE | yes | yes | yes |
| N | 3920 | 3918 | 3918 |
| adj. $R^2$ | 0.815 | 0.566 | 0.825 |

Note

*, **, *** represent significant at 10%, 5% and 1% significant levels, respectively. FE represent city fixed effects. Year means year fixed effects. Regression uses heteroscedasticity robust standard errors. Values below coefficients are standard errors. Constant term coefficient is not reported.

**Table 13. Producer services and consumer services.**

|  | (1) | (2) | (3) | (4) |
|---|---|---|---|---|
|  | lnpatent | lnps | lncs | lnpatent |
| lnps |  |  |  | 0.262*** |
|  |  |  |  | (0.093) |
| lncs |  |  |  | 0.401*** |
|  |  |  |  | (0.137) |
| pm25 | -0.596*** | -0.301*** | -0.328*** | -0.385** |
|  | (0.196) | (0.091) | (0.071) | (0.195) |
| Control | yes | yes | yes | yes |
| Year | yes | yes | yes | yes |
| FE | yes | yes | yes | yes |
| N | 3920 | 3918 | 3920 | 3918 |
| adj. $R^2$ | 0.815 | 0.481 | 0.556 | 0.822 |

Note

*, **, *** represent significant at 10%, 5% and 1% significant levels, respectively. FE represent city fixed effects. Year means year fixed effects. Regression uses heteroscedasticity robust standard errors. Values below coefficients are standard errors. Constant term coefficient is not reported.

variable, the coefficient symbols of the main regression are consistent with the previous text. The regression results in Table 12 show that the intermediary effect of the service industry is still robust. The intermediary effect model in Table 13 proves that haze affects both producer services and nonproducer services. Table 14 shows that advanced producer services have a stronger impact on innovation but are less affected by haze. After using other explained variables, the regression results are still robust.

## 5.3. Lag in explanatory variable

The explanatory variable applied in the previous paper is the current average concentration of urban $PM_{2.5}$. There are two possible hypotheses in theory: (1) the labor force chooses to

**Table 14. Advanced producer services and basic producer services.**

|  | (1) | (2) | (3) | (4) |
|---|---|---|---|---|
|  | lnpatent | lnaps | lnbps | lnpatent |
| lnaps |  |  |  | 0.364*** |
|  |  |  |  | (0.083) |
| lnbps |  |  |  | 0.193*** |
|  |  |  |  | (0.056) |
| pm25 | -0.596*** | -0.217*** | -0.357*** | -0.448** |
|  | (0.196) | (0.071) | (0.134) | (0.191) |
| Control | yes | yes | yes | yes |
| Year | yes | yes | yes | yes |
| FE | yes | yes | yes | yes |
| N | 3920 | 3918 | 3920 | 3918 |
| adj. $R^2$ | 0.815 | 0.481 | 0.556 | 0.822 |

Note

*, **, *** represent significant at 10%, 5% and 1% significant levels, respectively. FE represent city fixed effects. Year means year fixed effects. Regression uses heteroscedasticity robust standard errors. Values below coefficients are standard errors. Constant term coefficient is not reported.

**Table 15. Mechanism analysis of services.**

|  | (1) | (2) | (3) |
|---|---|---|---|
|  | **lninnov** | **lnse** | **lninnov** |
| lnse |  |  | 0.605*** |
|  |  |  | (0.105) |
| L.pm25 | -0.498*** | -0.292*** | -0.318** |
|  | (0.155) | (0.062) | (0.151) |
| Control | yes | yes | yes |
| Year | yes | yes | yes |
| FE | yes | yes | yes |
| $N$ | 3920 | 3918 | 3918 |
| adj. $R^2$ | 0.896 | 0.565 | 0.902 |

Note

*, **, *** represent significant at 10%, 5% and 1% significant levels, respectively. FE represent city fixed effects. Year means year fixed effects. Regression uses heteroscedasticity robust standard errors. Values below coefficients are standard errors. Constant term coefficient is not reported.

migrate according to the current air quality; and (2) the migration and industrial change occur within one year. Considering that the air quality of the current period may not immediately affect the economic activities of the current period, we refer to Chen and Chen (2018). In this section, the lag in PM$_{2.5}$ (L.pm25) is used as the core explanatory variable to test the robustness of the previous conclusions. After replacing the core explanatory variables, the results of the main regression are reported in Tables 15–17. The results are also consistent with the previous mechanistic analysis.

## 6. Conclusions

This paper employs the urban panel data of China from 2003 to 2016 as the research sample, uses the bidirectional fixed effect model and the intermediary effect model to investigate the

**Table 16. Producer services and consumer service.**

|  | (1) | (2) | (3) | (4) |
|---|---|---|---|---|
|  | **lninnov** | **lnps** | **lncs** | **lninnov** |
| lnps |  |  |  | 0.234*** |
|  |  |  |  | (0.084) |
| lncs |  |  |  | 0.416*** |
|  |  |  |  | (0.118) |
| L.pm25 | -0.498*** | -0.249*** | -0.318*** | -0.305** |
|  | (0.155) | (0.075) | (0.062) | (0.152) |
| Control | yes | yes | yes | yes |
| Year | yes | yes | yes | yes |
| FE | yes | yes | yes | yes |
| $N$ | 3920 | 3918 | 3920 | 3918 |
| adj. $R^2$ | 0.896 | 0.480 | 0.556 | 0.903 |

Note

*, **, *** represent significant at 10%, 5% and 1% significant levels, respectively. FE represent city fixed effects. Year means year fixed effects. Regression uses heteroscedasticity robust standard errors. Values below coefficients are standard errors. Constant term coefficient is not reported.

**Table 17. Advanced producer services and basic producer services.**

|  | (1) | (2) | (3) | (4) |
|---|---|---|---|---|
|  | lninnov | lnaps | lnbps | lninnov |
| lnaps |  |  |  | 0.342*** |
|  |  |  |  | (0.090) |
| lnbps |  |  |  | 0.186*** |
|  |  |  |  | (0.051) |
| L.pm25 | -0.005*** | -0.002*** | -0.003*** | -0.004** |
|  | (0.002) | (0.001) | (0.001) | (0.001) |
| Control | yes | yes | yes | yes |
| Year | yes | yes | yes | yes |
| FE | yes | yes | yes | yes |
| N | 3920 | 3918 | 3920 | 3918 |
| adj. $R^2$ | 0.896 | 0.603 | 0.278 | 0.902 |

Note

*, **, *** represent significant at 10%, 5% and 1% significant levels, respectively. FE represent city fixed effects. Year means year fixed effects. Regression uses heteroscedasticity robust standard errors. Values below coefficients are standard errors. Constant term coefficient is not reported.

influence of haze on innovation and its heterogeneity, and discusses the mechanism of haze influencing innovation through the service industry. The following main conclusions are obtained: (1) as a whole, haze significantly inhibits the level of urban innovation. The higher the haze concentration is, the lower the urban innovation ability is. (2) Subregional sample regressions show that urban innovation in the eastern region is most affected by haze, while the central region is less affected, and the western region is not affected. Innovation in key cities and resource-dependent cities is less affected by haze. (3) Impact mechanism analysis shows that haze affects urban innovation not by influencing agricultural and manufacturing industries but by services. Both productive and nonproducer services are affected by haze and then act on innovation. Further research on the cluster of producer services shows that the influence mechanism of advanced producer services differs from that of basic producer services. Basic producer services are more sensitive to haze than advanced producer services but contribute less to innovation.

## Supporting information

**S1 Data.**
(ZIP)

## Acknowledgments

This research received no external funding. The authors declare no conflict of interest.

## Author Contributions

**Conceptualization:** Yong You.

**Writing – original draft:** Yong You.

**Writing – review & editing:** Yong You.

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
