## [Decision Letter · Decision Letter 0]

28 Jun 2021

PONE-D-21-16052

Air pollution, service development and innovation: evidence from China

PLOS ONE

Dear Dr. 游,

Thank you for submitting your manuscript to PLOS ONE. After careful consideration, we feel that it has merit but does not fully meet PLOS ONE’s publication criteria as it currently stands. Therefore, we invite you to submit a revised version of the manuscript that addresses the points raised during the review process.

We look forward to receiving your revised manuscript.

Kind regards,

Ghaffar Ali, PhD

Academic Editor

PLOS ONE

Journal Requirements:

3. We note you have included a table to which you do not refer in the text of your manuscript. Please ensure that you refer to Table 1 in your text; if accepted, production will need this reference to link the reader to the Table.

Additional Editor Comments (if provided):

Reviewers' comments:

Reviewer's Responses to Questions

**Comments to the Author**

1. Is the manuscript technically sound, and do the data support the conclusions?

Reviewer #1: Yes

Reviewer #2: Yes

2. Has the statistical analysis been performed appropriately and rigorously? 

Reviewer #1: Yes

Reviewer #2: Yes

3. Have the authors made all data underlying the findings in their manuscript fully available?

Reviewer #1: No

Reviewer #2: Yes

4. Is the manuscript presented in an intelligible fashion and written in standard English?

Reviewer #1: Yes

Reviewer #2: Yes

5. Review Comments to the Author

Reviewer #1: The article explains the influence of PM2.5 on urban innovations in China. The assessed issue is valid.

However, I found the article extremely difficult to follow. There is plenty of information that could be comprised to deliver the message more effectively. Most of the tables can be moved to the supplement. They just add complexity to the main text.

Explanations for abbreviations are missing.

Definition of concepts are missing. :e.g., What is pollution paradise effect, staff following the enterprise, urban innovation index, mediation effect, etc.

Page2. Introduction:

Please explain the different concepts; “pollution paradise”, “enterprises following staff’, “staff following the environment”.

Line 49-52. The transition from agricultural systems to industrialization is not going worldwide at the same pace. Please revise the statement. Check :

https://www.sciencedirect.com/science/article/pii/S0040162517311526?via%3Dihub

Benchmark, model and variable selection:

Line 103: Mention seven dimensions -eight variables, but there are just six, as one ‘economic development level’ is twice.

Baseline model:

Line 171 – 173. If haze concentration is declining why haze will continue having the same negative impact on innovation?

Heterogeneity:

Line 204: Natural resources industry? Not sure this is the right term. Industries that rely heavily on natural resources as part of the production processes. Give examples of type of industries.

Line 214: What is high emission projects?

When you talk about non-resource dependent cities, resource dependent, key cities non-key cities. I suppose there are key-cities/ resource dependent and vice versa. How do you separate this? Or you include the cities twice?

Effects of industrial structure:

Please mention the types of industries. The findings (line-244 to 246) are interesting. However, I still believe that the ‘methods’ and tables should be moved somewhere else. In that way, the author can write a section about results and not put all together. It is very difficult to follow.

Conclusions:

The author concludes about environmental governance. Did you analyze this in the study? I did not see governance in the analysis. Then you cannot conclude about something is not mention in the paper.

Reviewer #2: This is innovative and robust research that employs the urban panel data of China as the research sample to investigate the influence of haze on innovation and its heterogeneity and discusses the mechanism of haze influencing innovation through the service industry. It is a well-written paper in which the author brings a relevant theme for discussion. However, some issues must be addressed, and the following comments are taken into consideration.

1 - Some parts of the text the relationship between environmental pollution and economic activities is lost, because the focus is on haze, so I recommend to characterized haze pollution and the relationship between environmental pollution and haze.

2 – Along with the text, commas are missing, I suggest revising all the text.

3 – The abbreviation PM2.5 must have the 2.5 in a subscripted letter.

Line 16, 29, 46 – the author should specify what kind of pollution he is referring to in the text, like air pollution instead of “pollution”. Same in the following text.

Line 26 – The term “production emissions” is not clear in the sentence to me.

Line 29 – I suggest removing the “etc” and finish the sentence with “increasing the incidence of diseases”.

Line 30 - I recommend a brief introduction about haze pollution before addressing its consequences.

Line 31 – Consider changing the expression used “on the other hand”, because both sentences are standing poor outcomes of haze pollution.

Line 34-76 Bring the state of the art and contributes to the problematization of the topic, but could be more concise, directly focusing on the issue.

Line 96-97 - In Equation 1, PM2.5 is used as the core explanatory variable to measure haze pollution. It is usual to include the fine particulate matter in this equation? Why other pollutants like coarse PM and gaseous pollutants are not used?

Line 103 – Authors said that seven dimensions and eight variables were controlled, but in this section of the text, only the seven dimensions are cited. I suggest changing the text or include the variables.

Line 104 – The scale of the city and City level are not established in the text, which criteria was used could be added.

Line 127 – 129 The annual mean levels of PM2.5 and meteorological parameters were used. If the differences among the seasons of the year were included, could it lead to a different result?

Line 128 -129 - In the study, the meteorological parameters of sunshine hours, precipitation, and average temperature were included. Other meteorological data as wind speed and humidity could influence the concentration of PM2.5 and haze formation. Did you consider using it?

Line 195 – In the discussion about the regions could be included the socioeconomic status of the places is a contributing factor since higher socioeconomic status is usually related to lower air pollution and higher rates of innovation.

Line 227 - The sentence “this paper confirms that the influence of haze on pollution varies with regional…” is confused, because the pollution is the one contributing to the haze formation, and both interfere in the innovation. So, I suggest making the sentence clearer.

Line 322 – The sentence “On the other hand, innovation and production efficiency improvement may affect pollution emissions and cause haze and innovation to have a two-way causality.” is not clear, consider changing.

6. PLOS authors have the option to publish the peer review history of their article (what does this mean?). If published, this will include your full peer review and any attached files.

Reviewer #1: No

Reviewer #2: **Yes: **Bruna Marmett

---

## [Author Response · Author response to Decision Letter 0]

16 Sep 2021

Thanks for the referee’s professional comments, which not only improve the quality of the article, but also provide a great inspiration for our future research. 

In response to referee 1, we explain our data source and then response every comment (in red). For referee 2, we re-write related contents and give detailed explanations from 3 respects.

Responses to referee 1

Referee 1 may hold the view that our data is not fully available. We give a supplementary explanation. Our data have three sources:

(1) The global PM2.5 satellite remote sensing data are from Socioeconomic Data and Applications Center (SEDAC). 

The download link is https://sedac.ciesin.columbia.edu/data/set/sdei-global-annual-gwr-pm2-5-modis-misr-seawifs-aod/data-download.

(2) The Yearbook of Urban Statistics of China are published from the National Bureau of Statistics of China. The information of the yearbook can be found at http://www.stats.gov.cn/tjsj/tjcbw/201907/t20190708_1674721.html. The yearbook is published annually and publicly available. It can be bought from the China Statistics Press which published the yearbook at least. 

There is also a digital version at https://data.cnki.net/yearbook/Single/N2021050059. For example, You can choose year like ”2020年” and download the excel data in the “下载” Column. The date is available for all China Universities. But I am not sure if it is available for all visitors.

(3) The innovation index is calculated by Kou and Liu (2017) which is available on the researcher’s website (https://kouzonglai.blog.caixin.com/archives/176063). In the website, the Kou and Liu provided the data in a cloud disk. 

The download link is (https://pan.baidu.com/s/1qZydmzU) where the password is vgk5. The innovation index file is named “城市创新指数2001—2016（中国城市和产业创新力报告2017—寇宗来、刘学悦）.xlsx”. 

Those source data are all available. At last, we also upload your minimal data set as a Supporting Information file. The data set is processed by Stata14.

Section 1

The article explains the influence of PM2.5 on urban innovations in China. The assessed issue is valid.

However, I found the article extremely difficult to follow. There is plenty of information that could be comprised to deliver the message more effectively. Most of the tables can be moved to the supplement. They just add complexity to the main text.

Explanations for abbreviations are missing.

Definition of concepts are missing. :e.g., What is pollution paradise effect, staff following the enterprise, urban innovation index, mediation effect, etc.

We are appreciated for the referee’s comments. We revise the subscription and give a supplementary explanation. The pollution haven hypothesis, or pollution paradise haven effect, is the idea that polluting industries will relocate to jurisdictions with less stringent environmental regulations. And the effect has been empirically tested by plenty of studies. Recent trade and environmental policy debates seem to take as given that regulatory stringency in developed countries shifts polluting industries to the developing world.（Levinson and Taylor, 2008）.

Workers-following-firms (Staff following the enterprise) and firms-following-workers are different. Generally speaking, in China, manufacturing industry is workers-following-firms where the workers are mobile while other factors (raw materials, transportation cost, production cost and market) are settled which even leads to a large-scale population migration. However, the IT service is less sensitive to those factors because the production can be easily connected through the Internet.. Amenities could attract workers and talents which give birth to the new enterprises. Therefore, amenities play an important role in regional attractiveness for IT service employment whereas their role is less important in IT manufacturing employ.

urban innovation index. The innovation index is based on patent renewal model which was proposed by Pake and Schankerman’s (1984) to evaluate patents’ value. Kou and Liu (2017) calculate the value of all the patents, and then add them up into cities. 

mediation effect. A mediation effect occurs when the third variable (mediator, M) carries the influence of a given predictor variable (or explanatory variable) X to a given response variable(or explained variable ) Y. According to Baron & Kenny (1986) and Wen and Ye(2014), the standard intermediary effect test has "three steps" as follows:

█(Y=cX+e_1#（1） )

█(M=aX+e_2#（2） )

█(Y=c^' X+bM+e_3#（3） )

In equation (1), c is the X’s total effect on Y. The coefficient a reflects the relationship between X and M. Equation 3 means that X’s effect on Y become c^' after controlling the mediator M. That means part of the total effect on Y is caused by mediator M, or M carries the influence of X to Y. Figure R1 represents the relationship.

Figure R1: The mediation effect

The mediation is usually used in Mechanism analysis. In regression model, with 2 conditions satisfied:

(1) The coefficients c and a are significant.

(2) The absolute value of c^' is smaller than that of c, or c^' is no longer significant.

That means part of the effect, or the whole effect of X is by means of M.

Page2. Introduction:

Please explain the different concepts; “pollution paradise”, “enterprises following staff’, “staff following the environment”.

Line 49-52. The transition from agricultural systems to industrialization is not going worldwide at the same pace. Please revise the statement. Check :

https://www.sciencedirect.com/science/article/pii/S0040162517311526?via%3Dihub

Line 103: Mention seven dimensions -eight variables, but there are just six, as one ‘economic development level’ is twice.

Thanks for the referee’s recommendation of correcting. We modify the related contents in the article.

Baseline model:

Line 171 – 173. If haze concentration is declining why haze will continue having the same negative impact on innovation?

Maybe we did not describe the result clearly which caused some misunderstanding. In regression models, the coefficient means how much the explained variable changes on average while the explanatory variable varies 1 unite after controlling other variables. For example, in our regression:

█(lninnov_it= θ_0+θ_1 pm25+θ_2 Control_it+ϵ_it #(1) )

With every unite’s change in pm25，lninnov varies θ_1 unite. θ_1<0 means that haze (pm25) has a negative effect on innovation. The higher the concentration, the less the innovation. The haze concentration is declining when haze has the same negative impact on innovation for the controlled group -- a city without haze.

Heterogeneity:

Line 204: Natural resources industry? Not sure this is the right term. Industries that rely heavily on natural resources as part of the production processes. Give examples of type of industries.

Line 214: What is high emission projects?

When you talk about non-resource dependent cities, resource dependent, key cities non-key cities. I suppose there are key-cities resource dependent and vice versa. How do you separate this? Or you include the cities twice?

Thanks to the referee’s advice. Natural resources industry means the industries related to Natural Resources and Mining. The resource-dependent cities in our research are related to those cities rich in oil, coal and ore. The list of resource-dependent cities is published by the State Council of China(http://www.gov.cn/zwgk/2013-12/03/content_2540070.htm), in which the resource cities are rich in natural resources as oil, coal, ore and wood. Considering that wood is a renewable resource, we exclude 3 forest cities from all the 66 resource cities in the list. Because of the advantage in resource, the mining and metallurgy(mining and washing of coal, processing of ferrous metalsores, processing of nonferrousmetals ores, oli processing and coking, etc.) in those cities are well developed. That’s why the resource-dependent cities high have emission projects

Logically, the reviewer's concerns about resource-dependent cities and key cities are indeed necessary. But in fact, resource-based cities do not coincide with key cities. There is no key-city which is resource cities. A resource-based city is rich in natural resources, like oil, coal, iron ore and so on. However the key cities are provincial capital cities and sub-provincial cities which usually have a long history and are not famous for mineral resources. 

For the purpose of distinguishing the impact of haze on innovation in different natural resources environment and different policy environment, we compared different groups. 

Effects of industrial structure:

Please mention the types of industries. The findings (line-244 to 246) are interesting. However, I still believe that the ‘methods’ and tables should be moved somewhere else. In that way, the author can write a section about results and not put all together. It is very difficult to follow.

The first industry usually includes agriculture, forestry and fishing. The Second industry includes mining and quarrying, manufacturing, public utilities and construction. The types of industries are based on International Standard Industrial Classification of All Economic Activities (https://unstats.un.org/unsd/classifications/Econ/isic) and China’s National Bureau of Statistics(http://www.stats.gov.cn/tjsj/tjbz/201301/t20130114_8675.html). We summarize the industry classification as follows. 

Table R1 The first, second and third industry 

Conclusions:

The author concludes about environmental governance. Did you analyze this in the study? I did not see governance in the analysis. Then you cannot conclude about something is not mention in the paper.

The reviewer’s suggestions are rigorous which effectively improve the quality of the article. We did over-extend our findings. In response, we delete the conclusions about environmental governance in the Conclusions Section.

References

[1] Oliveira J. Implementing environmental policies in developing countries through decentralization: the case of protected areas in bahia, brazil. World Development. 2002; 30: 1713-1736.

[2] Copeland B, Taylor M. Trade, Growth and the Environment. Journal of Economics Literature. 2004; 42: 7–71.

[3] Levinson A, Taylor M. Unmasking the pollution haven effect. International Economic Review. 2008; 49: 223-254.

[4] Kou Z, Liu X. FIND Report on City and Industrial Innovation in China (2017), Fudan Institute of Industrial Development, School of Economics, Fudan University. 2017.

Responses to referee 2

Thanks for the referee’s professional comments, which not only improve the quality of the article, but also provide a great inspiration for our future research. 

Before we response to the referee’s proposal, we give a supplementary explanation about our data first. Our data have three sources:

(1) The global PM2.5 satellite remote sensing data are from Socioeconomic Data and Applications Center (SEDAC). 

The download link is https://sedac.ciesin.columbia.edu/data/set/sdei-global-annual-gwr-pm2-5-modis-misr-seawifs-aod/data-download.

(2) The Yearbook of Urban Statistics of China are published from the National Bureau of Statistics of China. The information of the yearbook can be found at http://www.stats.gov.cn/tjsj/tjcbw/201907/t20190708_1674721.html. The yearbook is published annually and publicly available. It can be bought from the China Statistics Press which published the yearbook at least. 

There is also a digital version at https://data.cnki.net/yearbook/Single/N2021050059. For example, You can choose year like ”2020年” and download the excel data in the “下载” Column. The date is available for all China Universities. But I am not sure if it is available for all visitors.

(3) The innovation index is calculated by Kou and Liu (2017) which is available on the researcher’s website (https://kouzonglai.blog.caixin.com/archives/176063). In the website, the Kou and Liu provided the data in a cloud disk. 

The download link is (https://pan.baidu.com/s/1qZydmzU) where the password is vgk5. The innovation index file is named “城市创新指数2001—2016（中国城市和产业创新力报告2017—寇宗来、刘学悦）.xlsx”. 

Those source data are all available. At last, we also upload your minimal data set as a Supporting Information file. The data set is processed by Stata14.

In response to the referee’s proposal, we modify the article from three aspects: (1) content, (2) idea, (3) grammar.

# content

1 - Some parts of the text the relationship between environmental pollution and economic activities is lost, because the focus is on haze, so I recommend to characterized haze pollution and the relationship between environmental pollution and haze.

Line 30 - I recommend a brief introduction about haze pollution before addressing its consequences.

Line 34-76 Bring the state of the art and contributes to the problematization of the topic, but could be more concise, directly focusing on the issue.

 Line 16, 29, 46 – the author should specify what kind of pollution he is referring to in the text, like air pollution instead of “pollution”. Same in the following text.

Those comments pinpoint a main weakness for the content. In response, we re-write related sections, providing more detailed explanations and concise descriptions.

# idea

Line 96-97 - In Equation 1, PM2.5 is used as the core explanatory variable to measure haze pollution. It is usual to include the fine particulate matter in this equation? Why other pollutants like coarse PM and gaseous pollutants are not used?

It is a good question. In terms of logical completeness, we should use other air pollution indicators like coarse PM and gaseous pollutants to measure haze pollution. However, based on two reasons, we use PM2.5 rather than other indicators finally.

(1) PM2.5 has a more serious impact on health than coarse PM.

The difference between PM2.5 (particles less than 2.5 micrometers in diameter) and other coarse PM (PM10, PM100) is the diameter. It has been found that PMs with an aerodynamic diameter smaller than 10 µm have a greater impact on human health. PMs with an aerodynamic diameter bigger than 10 µm are usually sand and dust, which can not penetrate into the lung and trachea easily. So does most of PM10. Exposure to fine particulate matter (PM2.5) is associated with cardiopulmonary mortality, cerebrovascular events, and overall mortality. The principal source of PM2.5 is vehicle emissions and industrial emissions.

Thus, we pay more attention to PM2.5 as the choice of core explanatory variable, which can consequently penetrate deeply into the lung, irritate and corrode the alveolar wall, comparing to coarse PM.

(2) PM2.5 is available in date.

Due to their large aerodynamic size, coarse PM could not stay in the air for a long time while PM2.5 could stay suspended in the air. PM2.5 can scatter light at specific wavelengths for its specific diameter which can be easily detected by remote sensing satellites. The formation of PM2.5 is related to gaseous pollutants such as NOx emissions, NH3. Single gaseous pollutant can not describe the air condition while the PM2.5 is a comprehensive indicator. A lot of studies have used the satellite remote sensing data of global PM2.5 as mentioned in the article.

Finally, we use PM2.5 as our core explanatory variable according to the existed studies about China’s air pollution which is satellite remote sensing data of global PM2.5. The data is from Socioeconomic Data and Applications Center (SEDAC). http://ciesin.columbia.edu/data.

Line 104 – The scale of the city and City level are not established in the text, which criteria was used could be added.

I am sorry, but I could not get the idea. The scale of the city usually has two meanings: the population and the area. And the population and area are positively related in China. We have used lnpop (log of population) to control the impact of city sclae in the text. If we use lnarea (log of the city’s area) as the variable describing city’s scale, the result is similar as follows. 

Figure R2: Population and area are related in China

Table R2. Baseline model (with city area)

 (1) (2) (3) (4) (5)

 OLS OLS FE FE FE

pm25 -0.216*** -0.040 -1.378*** -0.646*** -2.635***

 (0.068) (0.066) (0.150) (0.158) (0.455)

(pm25)^2 1.792***

 (0.343)

lnfina 0.901*** 0.864*** 0.870*** 0.027 0.039

 (0.017) (0.023) (0.038) (0.081) (0.081)

lnarea 0.484*** 0.477*** 0.389*** 0.232*** 0.222***

 (0.032) (0.031) (0.083) (0.060) (0.058)

second 0.472*** 0.391*** 0.713*** 0.349** 0.321*

 (0.090) (0.096) (0.127) (0.174) (0.172)

urban -0.703*** -0.471*** 0.065 0.107 0.049

 (0.082) (0.079) (0.162) (0.124) (0.125)

lnfdi 0.012** 0.020*** -0.000 -0.006 -0.005

 (0.006) (0.005) (0.006) (0.006) (0.006)

lnpgdp 0.005 -0.043 -0.032 -0.269*** -0.256***

 (0.025) (0.028) (0.084) (0.067) (0.065)

net 0.870*** 0.708*** 1.320*** 0.685*** 0.662***

 (0.141) (0.113) (0.254) (0.192) (0.191)

lnroad 0.149*** 0.125*** 0.220*** 0.002 0.007

 (0.016) (0.016) (0.055) (0.046) (0.046)

Year no yes no yes yes

FE no no yes yes yes

N 3918 3918 3918 3918 3918

adj. R2 0.894 0.848 0.898 0.899 0.894

Line 127 – 129 The annual mean levels of PM2.5 and meteorological parameters were used. If the differences among the seasons of the year were included, could it lead to a different result?

We thank the referee for this suggestion. Maybe we didn't describe the data clearly. The PM2.5 is an annual data. We could not find the seasonal differences from the date despite of the pity.

Line 128 -129 - In the study, the meteorological parameters of sunshine hours, precipitation, and average temperature were included. Other meteorological data as wind speed and humidity could influence the concentration of PM2.5 and haze formation. Did you consider using it?

We absolutely approve the comments about the IVs. Theoretically, these variables are all related to the formation of haze and they are not related to the level of economic development, which means that they are ideal IVs. Considering the availability of data, we tried to use wind speed, sunshine hours, precipitation and average temperature as our IV at first. But technically, the other IVs could not be perfectly tested statistically and the assumption of strictly exogenity is not satisfied. So we have to choose precipitation among all variables as our IV. 

Line 195 – In the discussion about the regions could be included the socioeconomic status of the places is a contributing factor since higher socioeconomic status is usually related to lower air pollution and higher rates of innovation.

Yes, higher socioeconomic status is usually related to lower air pollution and higher rates of innovation. That is why we control some socioeconomic factors (population, GDP per capita GDP, urbanization rate, etc.) in the model. Then our discussion about different regions has excluded those socioeconomic factors and the regional heterogeneity is ascribed to other factors as industrial structure mentioned below.

#3 grammer

2 – Along with the text, commas are missing, I suggest revising all the text.

3 – The abbreviation PM2.5 must have the 2.5 in a subscripted letter.

Line 26 – The term “production emissions” is not clear in the sentence to me.

Line 29 – I suggest removing the “etc” and finish the sentence with “increasing the incidence of diseases”.

Line 31 – Consider changing the expression used “on the other hand”, because both sentences are standing poor outcomes of haze pollution.

Line 103 – Authors said that seven dimensions and eight variables were controlled, but in this section of the text, only the seven dimensions are cited. I suggest changing the text or include the variables.

Line 227 - The sentence “this paper confirms that the influence of haze on pollution varies with regional…” is confused, because the pollution is the one contributing to the haze formation, and both interfere in the innovation. So, I suggest making the sentence clearer.

Line 322 – The sentence “On the other hand, innovation and production efficiency improvement may affect pollution emissions and cause haze and innovation to have a two-way causality.” is not clear, consider changing.

According to the referee’s advice, we revise the text. Thanks again for the careful reading again.

References

[1] Mishra, G, Ghosh, K, Dwivedi, A, Kumar, M, Tripathi, S. An application of probability density function for the analysis of pm2.5 concentration during the covid-19 lockdown period. Science of The Total Environment. 2021; doi: 10.1016/j.scitotenv.2021.146681

[2] Tai A, Mickley, L, Jacob D. Correlations between fine particulate matter (pm2.5) and meteorological variables in the united states: implications for the sensitivity of pm2.5 to climate change. Atmospheric Environment. 2010; 44: 3976-3984. 

[3] Xing Y, Xu Y, Shi M, Lian Y. The impact of pm2.5 on the human respiratory system. Journal of Thoracic Disease. 2016; 8: E69-E74.

---

## [Decision Letter · Decision Letter 1]

8 Nov 2021

PONE-D-21-16052R1Air pollution, service development and innovation: evidence from ChinaPLOS ONE

Dear Dr. 游,

Thank you for submitting your manuscript to PLOS ONE. After careful consideration, we feel that it has merit but does not fully meet PLOS ONE’s publication criteria as it currently stands. Therefore, we invite you to submit a revised version of the manuscript that addresses the points raised during the review process.

Please pay attention to the comments of the reviewer. Edit the paper for English language by a native English speaker. Also revise your introduction and other sections carefully.

We look forward to receiving your revised manuscript.

Kind regards,

Ghaffar Ali, PhD

Academic Editor

PLOS ONE

Journal Requirements:

Reviewers' comments:

Reviewer's Responses to Questions

**Comments to the Author**

1. If the authors have adequately addressed your comments raised in a previous round of review and you feel that this manuscript is now acceptable for publication, you may indicate that here to bypass the “Comments to the Author” section, enter your conflict of interest statement in the “Confidential to Editor” section, and submit your "Accept" recommendation.

Reviewer #1: All comments have been addressed

Reviewer #2: All comments have been addressed

2. Is the manuscript technically sound, and do the data support the conclusions?

Reviewer #1: Yes

Reviewer #2: Yes

3. Has the statistical analysis been performed appropriately and rigorously? 

Reviewer #1: Yes

Reviewer #2: Yes

4. Have the authors made all data underlying the findings in their manuscript fully available?

Reviewer #1: Yes

Reviewer #2: Yes

5. Is the manuscript presented in an intelligible fashion and written in standard English?

Reviewer #1: Yes

Reviewer #2: Yes

6. Review Comments to the Author

Reviewer #1: Thanks a lot to the authors for taken on board the comments and improving the paper.

I see in some paragraphs 'pm2.5'. It should be corrected to PM2.5 (2.5 in subscript).

Reviewer #2: The authors performed the changes suggested in the first revision, however some minor revisions still needs to be done.

- Grammar check the title.

- The first sentence of the introduction is not clear.

- Lines 31 and 32 subscribe the size of the PM.

- Explain all the abbreviations in its first mention in the text.

- Revised the journal guidelines to the make sure the manuscript follows the guidelines.

7. PLOS authors have the option to publish the peer review history of their article (what does this mean?). If published, this will include your full peer review and any attached files.

Reviewer #1: No

Reviewer #2: No

---

## [Author Response · Author response to Decision Letter 1]

24 Dec 2021

Thank the editor and referees for the professional comments. It encourages me again. I feel sorry for my carelessness as subscript and so on. In this version, I revise my manuscript according to the comments. The details are as follows.

Responses to academic editor

Edit the paper for English language by a native English speaker. 

Also revise your introduction and other sections carefully.

Thanks for the editor’s advice. I recheck the paper and then find a native English speaker to improve the expression, especially in the Abstract and Introduction.

I also uploaded my figure in the Preflight Analysis and Conversion Engine (PACE) digital diagnostic. 

To satisfy the journal requirements, I rechecked and corrected the reference.

In Line 486 of “Revised Manuscript with Track Changes”.

Responses to referee 1

Thanks a lot to the authors for taken on board the comments and improving the paper.

I see in some paragraphs 'pm2.5'. It should be corrected to PM2.5 (2.5 in subscript).

Sorry for my carelessness. I subscribe the size of PM2.5 in Lines 31 and 32. There is some other ‘pm25’ for the variable is named ‘pm25’. Thanks again for referee’s careful examination.

Responses to referee 2

Thanks for the referee’s professional comments, which not only improve the quality of the article, but also provide a great inspiration for our future research.

- Grammar check the title.

I checked the grammar of the title and corrected in the manuscript.

- The first sentence of the introduction is not clear.

I modified the Introduction in the manuscript.

- Lines 31 and 32 subscribe the size of the PM.

I subscribed the size of PM2.5 in Lines 31 and 32 in the new version and I also reexamined the article. I also feel sorry for my carelessness.

- Explain all the abbreviations in its first mention in the text.

 I checked the text and finded the expression of "vote with their feet" in Line 64 and "resource trap" in Line 203 may be not clear. Line 64 and Line 203 are in the “Revised Manuscript with Track Changes”. I modified it.

- Revised the journal guidelines to the make sure the manuscript follows the guidelines.

Thanks to the referee’s advice. I have revised the journal guidelines. But I am not sure weather I have got the idea.

---

## [Decision Letter · Decision Letter 2]

31 Jan 2022

Air pollution, service development and innovation: evidence from China

PONE-D-21-16052R2

Dear Dr. 游,

We’re pleased to inform you that your manuscript has been judged scientifically suitable for publication and will be formally accepted for publication once it meets all outstanding technical requirements.

Kind regards,

Ghaffar Ali, PhD

Academic Editor

PLOS ONE

Additional Editor Comments (optional):

Reviewers' comments:

Reviewer's Responses to Questions

**Comments to the Author**

1. If the authors have adequately addressed your comments raised in a previous round of review and you feel that this manuscript is now acceptable for publication, you may indicate that here to bypass the “Comments to the Author” section, enter your conflict of interest statement in the “Confidential to Editor” section, and submit your "Accept" recommendation.

Reviewer #1: All comments have been addressed

2. Is the manuscript technically sound, and do the data support the conclusions?

Reviewer #1: Yes

3. Has the statistical analysis been performed appropriately and rigorously? 

Reviewer #1: Yes

4. Have the authors made all data underlying the findings in their manuscript fully available?

Reviewer #1: Yes

5. Is the manuscript presented in an intelligible fashion and written in standard English?

Reviewer #1: Yes

6. Review Comments to the Author

Reviewer #1: The authors have addressed the comments and improved the text. Now , it is clearer and the conclusions are easy to understand

7. PLOS authors have the option to publish the peer review history of their article (what does this mean?). If published, this will include your full peer review and any attached files.

Reviewer #1: No